# A System for Worldwide COVID-19 Information Aggregation

**Akiko Aizawa**[3] **Frederic Bergeron**[1] **Junjie Chen**[5] **Fei Cheng**[1] **Katsuhiko Hayashi**[5] **Kentaro Inui**[4]
**Hiroyoshi Ito**[6] **Daisuke Kawahara**[7] **Masaru Kitsuregawa**[3] **Hirokazu Kiyomaru**[1] **Masaki Kobayashi**[6]
**Takashi Kodama**[1] **Sadao Kurohashi**[1] **Qianying Liu**[1] **Masaki Matsubara**[6] **Yusuke Miyao**[5]
**Atsuyuki Morishima**[6] **Yugo Murawaki**[1] **Kazumasa Omura**[1] **Haiyue Song**[1] **Eiichiro Sumita**[2]
**Shinji Suzuki**[8] **Ribeka Tanaka**[1] **Yu Tanaka**[1] **Masashi Toyoda**[8]
**Nobuhiro Ueda**[1] **Honai Ueoka**[1] **Masao Utiyama**[2] **Ying Zhong**[6]

[1]Kyoto University  [2]NICT  [3]NII  [4]Tohoku University  [5]The University of Tokyo
[6]The University of Tsukuba  [7]Waseda University  [8]Institute of Industrial Science, the University of Tokyo

## Abstract

The global pandemic of COVID-19 has made the public pay close attention to related news, covering various domains, such as sanitation, treatment, and effects on education. Meanwhile, the COVID-19 condition is very different among the countries (e.g., policies and development of the epidemic), and thus citizens would be interested in news in foreign countries. We build a system for worldwide COVID-19 information aggregation [1] containing reliable articles from 10 regions in 7 languages sorted by topics. Our reliable COVID-19 related website dataset collected through crowdsourcing ensures the quality of the articles. A neural machine translation module translates articles in other languages into Japanese and English. A BERT-based topic-classifier trained on an article-topic pair dataset helps users find their interested information efficiently by putting articles into different categories.

## 1 Introduction

Due to the global COVID-19 epidemic and the rapid changes in the epidemic, citizens are highly interested in learning about the latest news, which covers various domains, including directly related news such as treatment and sanitation policies and also side effects on education, economy, and so on. Meanwhile, citizens would pay extra attention to global related news now, not only because the planet has been brought together by the pandemic, but also because they can learn from the news of other countries to obtain first-hand news. For example, the epidemic outbreak in Korea is one month earlier than in Japan. Japanese citizens

could prepare better for the epidemic if they had obtained more information from Korea. Citizens could learn from Asian countries about the effectiveness of masks before local official guidance. Universities can learn about how to arrange virtual courses from the experience of other countries. Thus, a citizen-friendly international news system with topic detection would be helpful.

There are three challenges for building such a system compared with systems focusing on one language and one topic (Dong et al., 2020; Thorlund et al., 2020):

- The reliability of news sources.

- Translation quality to the local language.

- Topic classification for efficient searching.

The interface and the construction process of the worldwide COVID-19 information aggregation system are shown in Figure 1. We first construct a robust multilingual reliable website collection solver via crowdsourcing with native workers for collecting reliable websites. We crawl news articles base on them and filter out the irrelevant. A high-quality machine translation system is then exploited to translate the articles into the local language (i.e., Japanese and English). The translated news are grouped into their corresponding topics by a BERT-based topic classifier. Our classifier achieves 0.84 F-score when classifying whether an article is about COVID-19 and substantially outperforms a keyword-based model by a large margin. In the end, all the translated and topic labeled news is demonstrated via a user-friendly web interface.

## 2 Methodology

We present the pipeline for building the worldwide COVID-19 information aggregation system, focusing on the three solutions to the challenges.

---

The authors are in alphabetical order.

[1]https://lotus.kuee.kyoto-u.ac.jp/NLPforCOVID-19/en

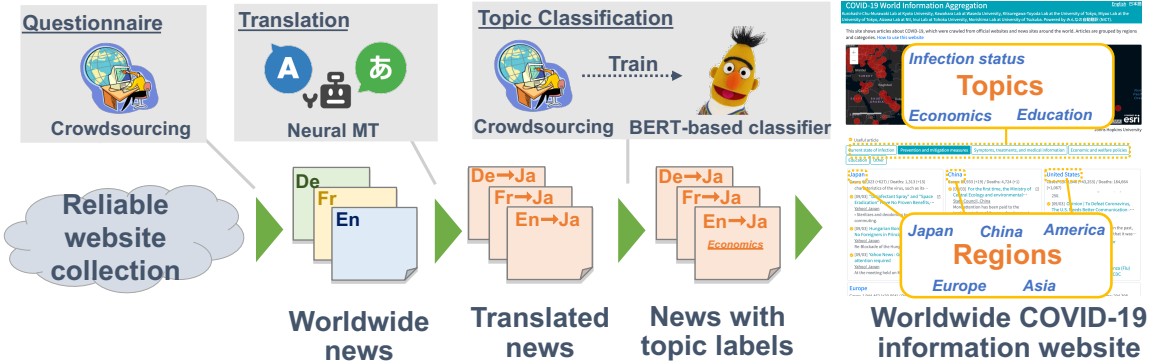

Figure 1: The construction process and the interface of the system.

| Website | Country | Primary | Reason | Topics |
|---|---|---|---|---|
| www.cdc.gov | US | True | The site is a government website, specifically the Center for Disease Control. | infection status prevention and emergency declaration symptoms, medical treatment and tests |
| www.covid19-yamanaka.com | Japan | False | Shinya Yamanaka is a famous medical researcher and his insights about COVID-19 are reliable. | prevention and emergency declaration |
| www.internazionale.it | Italy | False | This website collects and translates articles from news agencies and magazines from all over the world. Has up-to-date news, but also long-form analysis articles. Most of my deeper information comes from here. | infection status economics and welfare prevention and emergency declaration school and online classes |
| covid.saude.gov.br | Brazil | True | This site is the goverment web site. | infection status |

Table 1: Crowdworkers give trusted websites that they use to obtain COVID-19 related information. They also give the reasons to choose the website and what kind of information he/she obtaines from the website.

## 2.1 Reliable Website Collection

To avoid rumors and obtain high-quality, reliable information, it is essential to limit the information sources. Since we aim to create a multilingual system, the first challenge is to obtain a list of reliable information providers from different countries and in different languages.

Crowdsourcing is known to be efficient in creating high-quality datasets (Behnke et al., 2018). To collect the list of reliable websites of a specific country, we use multiple crowdsourcing services (e.g., Crowd4U[2], Amazon Mechanical Turk [3], Yahoo! Crowdsourcing[4], Tencent wenjuan[5]) and limit the workers' nationality because we assume that local citizens of each country know the reliable websites in their country. The workers not only suggest websites they think are reliable but they must also justify their choices and give a list of related topics they address, similar to constructing support for rumor detection (Gorrell et al., 2019; Derczynski et al., 2017).

We decided to use eight countries of interest, including India, the United States, Italy, Japan, Spain, France, Germany, and Brazil. For other countries or regions such as China and Korea, reliable websites are provided by international students from these areas.

We treat official news from the governments as primary information sources and reliable newspapers as secondary information sources. We counted how many times each website was mentioned by the crowdworkers and found that the primary information sources tend to be ranked at the top three in each country. So we mainly crawl articles from primary sources.

Table 1 shows examples of the crowdsourcing results. The workers provide websites indicating for each one whether it is a primary or a secondary source, what are the reasons to choose this particular website, and which topics are addressed by the website. These topics are selected from a list that includes eight topics (e.g., *Infection status*, *Economics and welfare*, *School and online classes*).

---

[2]https://crowd4u.org/
[3]https://www.mturk.com/
[4]https://crowdsourcing.yahoo.co.jp
[5]https://wj.qq.com

## 2.2 Crawl, Filter and Translation for Information Localization

We crawl articles from 35 most reliable websites everyday by accessing the entry page and jumping to urls inside it recursively.

The number of crawled web pages is too big and exceeds the translation capacity. We consider only the most relevant pages by filtering using keywords such as *COVID*. We can focus on pages with a higher probability to be COVID-19 related.

We use neural machine translation model Tex-Tra[6] with self-attention mechanism (Bahdanau et al., 2015; Vaswani et al., 2017). The translation system provides high-quality translation from news articles in multiple languages into articles in Japanese and English. The translation capacity is approximately 3,000 articles per day.

## 2.3 Topic Classification

To perform topic classification, we first collect the dataset via crowdsourcing. The topic labels are annotated to a subset of articles. Then we train a topic-classification model to label further articles automatically.

### 2.3.1 Crowdsourcing Annotation for Topic Classification

All articles are in Japanese and English after the translation stage, we then apply crowdsourcing annotation to label the articles with topics. As shown in Figure 2, the crowdsourcing workers first check the content of the page and give four labels to the article: whether it is related to COVID-19, whether it is helpful, whether the translated text is fluent, and topics of the article.

Each article is assigned to 10 crowdworkers from Yahoo Crowdsourcing and we set a threshold to $50\%$ for each binary question, i.e., if more than 5 workers think the article is related to COVID-19, then we label the article as $related$. We post this crowdsourcing task twice a week and can obtain 20K article-topic pairs each time.

### 2.3.2 Automatic Topic Classifier

The pretrained language model BERT (Devlin et al., 2019) shows reliable performance on many NLP tasks with limited annotated data including document classification (Adhikari et al., 2019; Sun et al., 2019). We use a pretrained BERT model in a feature based manner (Lee et al., 2019) where

⁶https://mt-auto-minhon-mlt.ucri.jgn-x.jp/

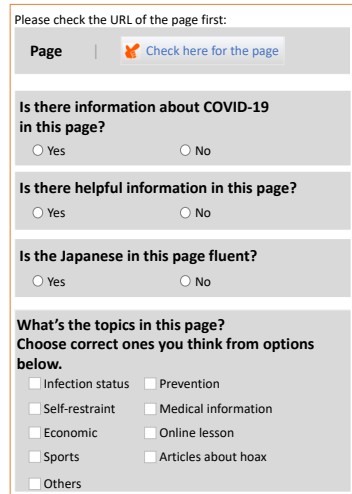

Figure 2: A sample of crowdsourcing annotation.

| Country \ # | Questionnaire | Reliable sites |
|---|---|---|
| India | 122 | 67 |
| US | 106 | 77 |
| Italy | 104 | 68 |
| Japan | 102 | 49 |
| Spain | 126 | 90 |
| France | 127 | 71 |
| Germany | 106 | 61 |
| Brazil | 115 | 67 |
| Total | 908 | 550 |

Table 2: Statistics of the number of questionnaires and reliable sites of each country.

| Country | Article with topic label |
|---|---|
| France | 31K |
| America | 6K |
| Japan | 5K |
| China | 8K |
| International | 3K |
| Spain | 1K |
| India | 2K |
| Germany | 1K |
| Total | 57K |

Table 3: Statistics of the article-topic dataset constructed by crowdsourcing.

encoder weights kept frozen and train a classifier using the labeled articles by crowdsourcing. The BERT-based topic classification can then label other pages.

We also compare it with a keyword-based baseline method where we set keywords for each topic and find exact match.

| Task | Keyword-based model | | | BERT-based model | | |
|------|---------------------|---|---|------------------|---|---|
| | Precision | Recall | F-score | Precision | Recall | F-score |
| Is about COVID-19 | 0.36 | 1.00 | 0.54 | 0.82 | 0.87 | **0.84** |
| Topic: Infection status | 0.09 | 0.53 | 0.16 | 0.43 | 0.81 | **0.56** |
| Topic: Prevention | 0.05 | 0.73 | 0.10 | 0.19 | 0.73 | **0.30** |
| Topic: Medical information | 0.17 | 0.70 | 0.27 | 0.27 | 0.91 | **0.41** |
| Topic: Economic | 0.06 | 0.36 | 0.10 | 0.14 | 0.84 | **0.24** |
| Topic: Education | 0.06 | 1.00 | **0.11** | 0.05 | 0.60 | 0.09 |
| Topic: Art and Sport | 0.06 | 0.41 | 0.10 | 0.08 | 0.94 | **0.14** |
| Topic: Others | 0.52 | 0.07 | 0.13 | 0.87 | 0.79 | **0.83** |

Table 4: Topic classification results. Each line stands for one task. We use F-score to evaluate.

| Topic | Positive | Negative | Positive percentage |
|-------|----------|----------|---------------------|
| Is about COVID-19 | 24361 | 32265 | 43.0% |
| Topic: Infection status | 6664 | 49962 | 11.8% |
| Topic: Prevention | 2533 | 54093 | 4.5% |
| Topic: Medical information | 5075 | 51551 | 9.0% |
| Topic: Economic | 2066 | 54560 | 3.6% |
| Topic: Education | 173 | 56453 | 0.3% |
| Topic: Art and Sport | 657 | 55969 | 1.2% |
| Topic: Others | 37331 | 19295 | 65.9% |

Table 5: The number of positive and negative samples for each topic.

## 3 Results

We show the statistics of the reliable website collection , topic classification results, translation quality evaluation and statistical information of the interface in this section.

### 3.1 Reliable Website Collection

As shown in Table 2, we totally recieved 908 questionnaire results from 8 countries with totally 550 websites. Rumors are rampant in this era, the reliable websites dataset can help people to protect themselves from COVID-19 and avoid trusting rumors about COVID-19.

### 3.2 Topic Classification

We compared the BERT-based model with the keyword-based baseline model on topic classification task.

For the keyword-based method, there are totally 76 selected keywords of different topics such as *COVID*, *Remote work*, and *Social distance*.

For the BERT-based method, we use the pre-trained BERT-LARGE model with Whole Word Masking (WWM). We add one linear layer after

| Topic | Krippendorff's alpha |
|-------|----------------------|
| Is about COVID-19 | 0.927 |
| Topic: Infection status | 0.898 |
| Topic: Prevention | 0.851 |
| Topic: Medical information | 0.867 |
| Topic: Economic | 0.931 |
| Topic: Education | 0.938 |
| Topic: Art and Sport | 0.994 |
| Topic: Others | 0.884 |

Table 6: Inter-annotator agreement measured by Krippendorff's alpha of each topic.

the BERT encoder without fine-tuning the encoder. For every article, we take the hidden state of the ending symbol of each sentence as the sentence embedding and perform mean and max pooling of all sentence embeddings. The input of the linear layer is the concatenation of mean and max pooling embeddings and the output is a binary label.

The article-topic dataset is shown in Table 3, it contains totally 57K articles from 8 countires

with topic label. We calculated the Krippendorff's alpha [7] as an inter-annotator agreement measure for 57K human checked articles, as shown in Table 6, the inter-annotator agreement of all topics is larger than 0.8 which is high enough (Krippendorff, 2004), showing the quality of our dataset is guaranteed.

We randomly selected select 90% data as train set and the remaining 10% as test set. As shown in Table 4, the BERT-based model outperforms the baseline model in almost all tasks. We can see that our system can reliably classify which articles are related to COVID-19 with 0.84 F1, and that our interface can show related news to our users. The topic classifiers achieve relatively satisfactory performance on the medical related topics (i.e. Infecction status, Prevention and Medical information). The general high recall guarantees the most topic relevant articles are returned. When presenting to end users, the topic predictions are filtered by the 'Is about COVID-19' classifier to ensure the eventual precision. Meanwhile, for some topics such as Arts & Sports and Education, the performance of the current system is still limited.

We further analyze the balance of the dataset by calculating the positive and negative number of each topic. As shown in Table 5, labels of several topics are imbalanced, for example, Education, Art and Sport, Economic. Analyzed together with the BERT topic classifier results, we found that the performance is poor for such imbalanced topics. For more frequent and balanced topics such as Infection status and Medical information, the F-scores are relatively higher.

### 3.3 Machine Translation Evaluation

For the evaluation of the translated text, we conducted human evaluation through crowdsourcing. As shown in Table 7, 61.7% of the articles are fluent in the translated language and the inter-annotator agreement is high enough (>0.8).

| Fluent | Not fluent | Krippendorff's alpha |
|--------|-----------|---------------------|
| 15036  | 9235      | 0.877               |

Table 7: Human evaluation of translated Japanese articles related to COVID-19 and inter-annotator agreement.

---

[7] https://github.com/pln-fing-udelar/fast-krippendorff

### 3.4 Statistics of the System

| Country | Raw(↑/day) | Translated | With topics |
|---------|-----------|-----------|-------------|
| France  | 774K(8K)  | 74K       | 9K          |
| US      | 69K(730)  | 15K       | 2K          |
| Japan   | 25K(260)  | 5K        | 2K          |
| Europe  | 50K(510)  | 2K        | 50          |
| China   | 38K(400)  | 3K        | 342         |
| Int.    | 45K(470)  | 3K        | 263         |
| Korea   | 16K(170)  | 260       | 71          |
| Spain   | 4K(40)    | 370       | 36          |
| India   | 14K(150)  | 860       | 66          |
| Germany | 16K(170)  | 8K        | 6K          |
| Total   | 1.05M(11K) | 110K     | 18K         |

Table 8: Statistics of the growing database of the system.

The detail of the system database is shown in Table 8. There are totally 1.05M website pages with 110K of them translated into Japanese and 18K of them with topic labels. The dataset is still growing approximately 11K pages per day.

We collected the number of visits to the website through google analytics, there are about 200 to 500 visits per month and more than 100 people at the peak per day. It suggests that the system is actually taken up by the public.

## 4 Conclusion

We built a system for worldwide COVID-19 information aggregation by combining crowdsourcing, crawling, machine translation, and a topic classifier, which provides reliable, comprehensive and latest information from the world. In the meanwile, we proposed an effective approach to annotate large cross-lingual news topic datasets with high inner-annotation agreement, which potentially benefits the NLP community to enrich the solutions for preventing COVID-19. The contextual BERT based classification models achieve reasonable performance considering the imbalance of the topic labels. We assume this work could attract future research interests to the COVID-19 related tasks.

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

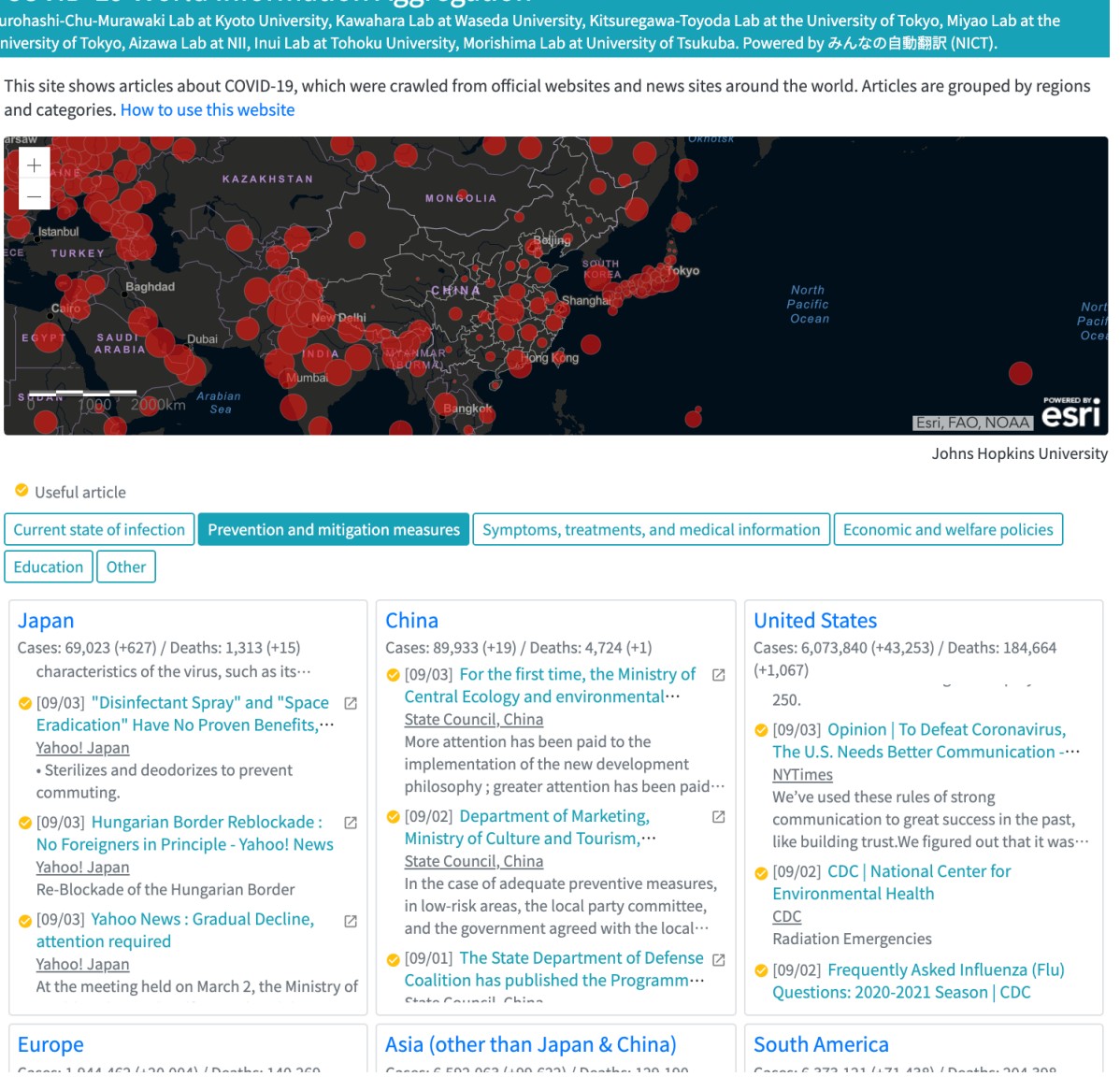

Figure 3: The interface of the worldwide COVID-19 information aggregation system in English.

| Website | Country | Primary | Reason | Topics |
|---|---|---|---|---|
| www.washingtonpost.com/coronavirus | US | True | I visit this URL daily and I trust them. | prevention and emergency declaration
symptoms, medical treatment and tests
economics and welfare |
| www.osha.gov/SLTC/covid-19 | US | False | It helps employees and employers understand the work environment better as far a covid19 goes and how to stay healthy, safe and follow guidleines correctly. | prevention and emergency declaration
symptoms, medical treatment and tests
economics and welfare |
| www.mhlw.go.jp/stf/seisakunitsuite
/bunya/0000164708_00001.html | Japan | True | The government provides reliable information. | infection status
prevention and emergency declaration |
| vdata.nikkei.com/newsgraphics
/coronavirus-world-map | Japan | False | I can learn the worldwide information through it. | infection status |
| www.ncbi.nlm.nih.gov/pubmed | Italy | True | A scientific papers site | symptoms, medical treatment and tests |
| www.ansa.it
/canale_saluteebenessere/ | Italy | False | This is an official news outlet that gathers its news from trusted sources, it's a constantly updated website which a lot of Italians rely on. It also has exclusive reports and interviews to important people. | infection status
prevention and emergency declaration
symptoms, medical treatment and tests
school and online classes
about rumours |
| www.gouvernement.fr
/info-coronavirus | France | True | This is French government website. | prevention and emergency declaration
symptoms, medical treatment and tests |
| aatishb.com/covidtrends | France | False | The code is open source and the data come from reliable source | infection status |
| cnecovid.isciii.es | Spain | True | This site is the goverment web site | infection status
prevention and emergency declaration
symptoms, medical treatment and tests |
| www.marca.com/tiramillas
/actualidad/2020/05/14/
5ebcc0cee2704ec4bb8b4623.html | Spain | False | It is a sport magazine but they constantly update all the information in Spain about coronavirus | infection status
entertainment and sports |
| www.charite.de | Germany | True | its the page of the hospital that mainly works with the german goverment | infection status |
| www.spiegel.de
/thema/coronavirus/ | Germany | False | one of Germanys oldest weekly news Paper, quality journalism and fact checking | infection status
economics and welfare
entertainment and sports |
| coronavirus.curitiba.pr.gov.br | Brazil | True | This is my city's COVID page, with daily updates on infection status and the running of the city, I can trust them because they only relay official information. | infection status
prevention and emergency declaration |
| www.uol.com.br | Brazil | False | Its the biggest news site here in my country | prevention and emergency declaration
entertainment and sports |
| www.mohfw.gov.in | India | True | This is the official page of the ministry of health and family welfare of government of India and is therefore reliable. | infection status
prevention and emergency declaration
symptoms, medical treatment and tests
economics and welfare
school and online classes
entertainment and sports |
| www.thehindu.com | India | False | This is one of the trusted News paper | infection status
prevention and emergency declaration
economics and welfare
school and online classes |

Table 9: Some crowdsourcing results of reliable websites collection

| Country | Website | Mentioned times |
|---|---|---|
| United States | www.cdc.gov/coronavirus/2019-ncov
www.usa.gov/coronavirus
www.nytimes.com/news-event/coronavirus | 14
6
4 |
| Japan | hazard.yahoo.co.jp/article/20200207
www.mhlw.go.jp/...
corona.feedal.com | 17
13
6 |
| Italy | www.salute.gov.it/nuovocoronavirus
www.salute.gov.it/portale/home.html
www.worldometers.info/coronavirus | 11
4
3 |
| France | www.gouvernement.fr/info-coronavirus
www.who.int/fr/emergencies/diseases/novel-coronavirus-2019
www.lemonde.fr/coronavirus-2019-ncov/ | 28
7
6 |
| Spain | www.usa.gov/coronavirus
www.mscbs.gob.es/profesionales/...
covid19.gob.es | 9
7
4 |
| Germany | www.rki.de/DE/Home/homepage_node.html
www.bundesgesundheitsministerium.de/coronavirus.html
interaktiv.morgenpost.de/corona-virus-karte-infektionen-deutschland-weltweit | 7
6
5 |
| Brazil | covid.saude.gov.br
g1.globo.com/bemestar/coronavirus
coronavirus.saude.gov.br | 21
11
9 |
| India | www.worldometers.info/coronavirus
www.mohfw.gov.in
www.mygov.in/covid-19/?cbps=1 | 11
10
10 |

Table 10: Top three mentioned websites by crowdworkers of each country.