# OpenReview forum: "A System for Worldwide COVID-19 Information Aggregation"
_EMNLP/2020/Workshop/NLP-COVID — NLP-COVID19-EMNLP Oral_

### Official Review · AnonReviewer1 · 2020-09-11
**Potentially beneficial system, but with little evaluation.**

**Rating:** 4
**Confidence:** 4

**Review:**

The paper describes a system that aggregates news and official sources on Covid-19 and presents them with a web interface to the public, as a way of providing reliable, comprehensive and up-to-date information. The system is international, making extensive use of machine translation. It makes use of crowdsourcing in two places. The first use is to solicit recommendations for good information sources. The second is to gather training and test data on individual articles, asking about relevance to Covid-19, helpfulness, the quality of translation and membership of 9 topics. The system also makes use of NLP components. A classifier, based on BERT, classifies articles as to whether they are relevant to Covid-19, and assigns them to the 9 topics. Furthermore, they deploy a pre-existing machine-translation system.

Reasons to accept: The system being described potentially meets a need for citizens to have access to reliable, relevant and up-to-date information; the value of this paper largely depends on whether the system lives up to this promise. Also, the system is of some interest to the NLP community. The article classifier appears to get good results for relevance-to-Covid-19 - an F score on 0.84, substantially above a keyword-based baseline. The list of reliable information sources that they have collected is
Furthermore it is often interesting to hear about the contexts into which NLP systems are to be deployed.

Reasons to reject:  In general this paper suffers from a lack of evaluation, or even thought about evaluation, except for the NLP components (the evaluation of the classifiers is OK and the machine translation is done using a published system). The introduction sets out a rationale for building such a system, but does not present any evidence that such a system would be taken up by the public, or result in them being better informed that without it - the benefits are simply asserted. The conclusion is very brief and makes no mention of how and even whether the impact of the system will be studied. I also have concerns about the reliability of the crowdsourced annotations. The system collects information from 10 people per article, yet no mention is made of how often the annotators agree. Relatedly, the F scores for classification into 9 topics are a lot lower than for the relevance-to-Covid-19 classifier. It is not clear whether this is caused by the classifier not being so good at the task, or whether the low scores reflect a task that is hard for humans to do consistenly. Some discussion of inter-annotator agreement would be helpful here.

Conclusion: This paper describes a system that has a lot of promise, but provides very little information about how well it lives up to its promise. As such I cannot recommend that the paper is published in its current state.

---

> ### Author Response · Authors · 2020-09-27
> **Thank you for your comments. We added evaluation of the dataset, the MT model,  and the public impact.**
>
> Thank you very much for your careful comments as well as thoughtful questions.
>
> **In general this paper suffers from a lack of evaluation. Relatedly, the F scores for classification into 9 topics are a lot lower than for the relevance-to-COVID-19 classifier. It is not clear whether this is caused by the classifier not being so good at the task, or whether the low scores reflect a task that is hard for humans to do consistently. Some discussion of the inter-annotator agreement would be helpful here.**
>
> We analyzed the inter-annotator agreement, the imbalance of labels of each topic, and conducted a human evaluation of the machine translation quality.
> Please refer to Table 1, 2, and 3 in the response to reviewer 2.
>
> As studied in Table 1 and 2, the inter-annotator agreement is high but the dataset is imbalanced for some labels. These explained why the classifiers performed poorly on the labels ‘Education’, ‘Art and Sport’. We will try to use resampling techniques to fine-tune the BERT model to alleviate the imbalance of the dataset in the camera-ready version.
>
> **does not present any evidence that such a system would be taken up by the public**
>
> Here is some evidence suggesting that the system is actually taken up by the public.
> We collected the number of visits to the website through google analytics.
>
> - Per day
>     - 10 ~ 20 people on average
>     - More than 100 people at the peak
> - Per month
>     - 200 ~ 500 people

---

### Official Review · AnonReviewer3 · 2020-09-11
**Important system but could benefit from more science and more rigorous evaluation**

**Rating:** 4
**Confidence:** 3

**Review:**

This paper addresses an important topic. I visited the website and it certainly seems like a complicated and useful tool.

I would really like to accept this paper based on the magnitude of work that the authors have put into the pipeline and dataset, but the paper in its current form seems a bit lacking in analysis and evaluation. A couple of notes:

How is this paper beneficial to the NLP community? It certainly is a useful website for the general public, but it is unclear to me what I have learned and what ideas can be useful for NLP researchers or others working in the area.

What are the details for the topic classification? E.g., what is the distribution of topics and how were they annotated, and with what annotator agreement? Did some articles belong to other topics? What was the ground truth? It could be helpful to show a confusion matrix for the BERT classifier.

Is there any evaluation of the translated text?

The main area of improvement that I see in this paper is that there is not much analysis of such a large dataset. Some potential ideas:
Are the topics that are annotated with the articles the same or similar to articles that could be discovered by unsupervised clustering methods? Did they vary for the countries based on the number of cases at the time?
Any possible analysis on the distribution of n-grams?

If there is a more thorough analysis of the system and/or an argument for what can be learned from this paper or how this paper can be useful for the NLP community, I would be happy to change my rating.

---

> ### Author Response · Authors · 2020-09-27
> **Thank you for your comments. We added evaluation of the dataset and the MT model.**
>
> Thank you very much for your careful comments as well as thoughtful questions.
>
> **How is this paper beneficial to the NLP community? It certainly is a useful website for the general public, but it is unclear to me what I have learned and what ideas can be useful for NLP researchers or others working in the area. What is the distribution of topics and how were they annotated, and with what annotator agreement?**
>
> Contribution to the NLP community: We proposed an effective approach to annotate large cross-lingual news topic datasets with a high inner-annotation agreement, which benefits the NLP community to enrich the solutions for preventing COVID-19. The classification models are based on the latest contextualized encoder BERT, which provides reasonable performance considering the imbalance of the topic labels. We believe this work will potentially attract future research efforts to challenge this task.
>
> We analyzed the inter-annotator agreement, the imbalance of labels of each topic, and conducted a human evaluation of the machine translation quality.
> Please refer to Table 1, 2, and 3 in the response of reviewer 2.
>
> **What are the details for the topic classification? Did some articles belong to other topics?**
>
> | # of articles related to COVID-19 | # of articles related to COVID-19 and contains Topic: Others | # of articles related to COVID-19 and contains Topic: Others only |
> | --------------------------------- | ------------------------------------------------------------ | ----------------------------------------------------------------- |
> | 24361                             | 7751                                                         | 6990                                                              |
>
> Table 4: Articles contains Topic: Others.
>
> Some articles belong to topics other than our designed topics. As shown in Table 4, there are 6990 articles containing only Topic: Other, out of 24361 COVID-19 related articles. It suggests that we may need to design other frequent topics for these articles.
>
> We public the code of topic classification here, which contains model details and keyword list.  https://github.com/NLPforCOVID-19/text-classifier
>
> **how were they annotated**
>
> Also, here is the instruction for annotators.
> ```
> Please check the web page related to COVID-19 and answer four questions.
>
> Instruction:
> 1. Click the 'check the web page' button and read the main text.
> 2. After reading the page, return to yahoo! crowdsourcing and answer the four questions.
> 3. Do steps 1 and 2 five times.
>
> Note:
> You only need to check the main text, there is no need to check the links to other pages.
> The page is translated to Japanese by machine translation where the original page is not in Japanese.
> The data is for academic purposes.
> ```
> We will add a more detailed evaluation in the camera-ready version.

---

### Official Review · AnonReviewer2 · 2020-09-25
**Large amount of COVID-19-related web-crawled data with human annotation. The evaluation and the usefulness of the topic classifier is less convincing.**

**Rating:** 5
**Confidence:** 3

**Review:**

The paper describes the construction of a website (system) that provides users with COVID-19-related information aggregated and translated from multiple reliable sources. Extensive use of crowdsourcing is done to check the reliability of the resources as well as to annotate them. The resources in languages other than English and Japanese are translated using a machine translation system called TexTra.

Reasons to accept: A large amount of human-annotated web documents is a valuable resource on its own. It is both costly and labor-consuming to build such a dataset.

Reason to reject: I am not sure if the evaluation of the topic classification is fair. A keyword-based baseline sounds too weak in comparison with a BERT-based neural network. A more sensible baseline would be an embedding-based similarity measure.
Also, even with the BERT-based topic classifier, the precision of some topics is very low. It would have been interesting to see if the inter-annotator agreement is also low for such topics. I think this is especially important because, in order to justify the validity of the dataset, the authors should have posted the reliability of the annotations among the annotators.
In addition, as the system makes heavy use of machine translation, a simple result table about the quality of the translation would be have been informative.

All in all, I acknowledge that the authors went to great lengths to construct a potentially useful dataset. However, its potential value of the dataset is not convincingly presented in the paper in its current state.

---

> ### Author Response · Authors · 2020-09-27
> **Thank you for your comments. We added evaluation of the dataset and the MT model.**
>
> Thank you very much for your careful comments as well as thoughtful questions.
>
> To provide a better insight into the dataset, we
> - analyzed the inter-annotator agreement,
> - analyzed the imbalance of labels of each topic and
> - conducted a human evaluation of machine translation quality.
>
> **It would have been interesting to see if the inter-annotator agreement is also low for such topics.**
>
> We calculated the Krippendorff's alpha (https://github.com/pln-fing-udelar/fast-krippendorff) as an inter-annotator agreement measure for 56626 human checked articles.
>
> | Topic                      | Krippendorff's alpha |
> | -------------------------- | -------------------- |
> | Is about COVID-19          | 0.927                |
> | Topic: Infection status    | 0.898                |
> | Topic: Prevention          | 0.851                |
> | Topic: Medical information&nbsp;&nbsp; | 0.867                |
> | Topic: Economic            | 0.931                |
> | Topic: Education           | 0.938                |
> | Topic: Art and Sport       | 0.994                |
> | Topic: Others              | 0.884                |
>
> Table 1: Krippendorff's alpha of all topics.
>
> It shows the inter-annotator agreement of all topics is larger than 0.8 which is high enough (Krippendorf, 2004),  meaning the quality of our dataset is guaranteed.
>
> **even with the BERT-based topic classifier, the precision of some topics is very low.**
>
> We further analyze the imbalance of the labels of each topic.
>
> | Topic                      | # of articles with true label&nbsp;&nbsp; | # of articles with false label&nbsp;&nbsp; | True percentage |
> | -------------------------- | ----------------------------- | ------------------------------ | --------------- |
> | Is about COVID-19          | 24361                         | 32265                          | 43.0%           |
> | Topic: Infection status    | 6664                          | 49962                          | 11.8%           |
> | Topic: Prevention          | 2533                          | 54093                          | 4.5%            |
> | Topic: Medical information&nbsp;&nbsp; | 5075                          | 51551                          | 9.0%            |
> | Topic: Economic            | 2066                          | 54560                          | 3.6%            |
> | Topic: Education           | 173                           | 56453                          | 0.3%            |
> | Topic: Art and Sport       | 657                           | 55969                          | 1.2%            |
> | Topic: Others              | 37331                         | 19295                          | 65.9%           |
>
> Table 2: Labels for 56626 human checked articles. “Topic: Others” contains most articles that are not COVID-19 related.
>
> As shown in Table 2, labels of several topics are imbalanced, for example, Education, Art and Sport, Economic. Analyzed together with the BERT topic classifier results, we found that the performance is poor for such imbalanced topics. For more frequent and balanced topics such as Infection status and Medical information, the F-scores are relatively higher.
>
> We will try to use resampling techniques to fine-tune the BERT model to alleviate the imbalance of the dataset in the camera-ready version.
>
> **In addition, as the system makes heavy use of machine translation, a simple result table about the quality of the translation would have been informative.**
>
> |  Fluent&nbsp;&nbsp;  | Not fluent&nbsp;&nbsp; | Fluent percentage&nbsp;&nbsp; | Krippendorff's alpha |
> | ------ | ---------- | ----------------- | -------------------- |
> | 15036  | 9235      | 61.7%             | 0.877                |
>
> Table 3: Human evaluation of translated Japanese articles related to COVID-19 and inter-annotator agreement.
>
> For the evaluation of the translated text, we conducted human evaluation through crowdsourcing. As shown in Table 3, about 60% of the articles are fluent in the translated language. The inter-annotator agreement is high enough (>0.8).
> One assumption is that the NMT model we use does only sentence-level translation rather than document-level translation.